# Learning chemical sensitivity reveals mechanisms of cellular response
William Connell [1,2,3], Kristle Garcia [3,4,5,6], Hani Goodarzi [3,4,5,6] & Michael J. Keiser [1,2,3] ✉

Chemical probes interrogate disease mechanisms at the molecular level by linking genetic changes to observable traits. However, comprehensive chemical screens in diverse biological models are impractical. To address this challenge, we develop ChemProbe, a model that predicts cellular sensitivity to hundreds of molecular probes and drugs by learning to combine transcriptomes and chemical structures. Using ChemProbe, we infer the chemical sensitivity of cancer cell lines and tumor samples and analyze how the model makes predictions. We retrospectively evaluate drug response predictions for precision breast cancer treatment and prospectively validate chemical sensitivity predictions in new cellular models, including a genetically modified cell line. Our model interpretation analysis identifies transcriptome features reflecting compound targets and protein network modules, identifying genes that drive ferroptosis. ChemProbe is an interpretable in silico screening tool that allows researchers to measure cellular response to diverse compounds, facilitating research into molecular mechanisms of chemical sensitivity.

Chemical probes are highly potent small molecules that selectively target known mechanism-of-action proteins[1]. These tools are crucial for understanding the role of specific proteins in biological processes and diseases, and have been instrumental in investigating a range of functions such as those related to the cell cytoskeleton, immunosuppression, mTOR signaling, protein kinase dynamics, and have often served as the starting point for drug development[1–3]. In addition to their primary use as therapeutic agents, drugs can serve as chemical probes in complex diseases like cancer. Addressing cancer heterogeneity necessitates precision clinical treatment strategies and research into the mechanisms that control disease resistance and sensitivity[4,5]. By improving our understanding of gene expression patterns contributing to variance in drug response, we can develop better solutions for cancer patients exploiting specific tumor vulnerabilities.

Ideally, we could test large libraries of chemicals on disease models, engineered cell lines, and patient samples to probe disease mechanisms. However, screening biological samples against a large library of chemical probes is resource-prohibitive. To overcome this problem, a variety of traditional machine-learning methods have been applied to predict drug response, including support vector machines (SVMs), random forests (RFs), and multi-layer perceptrons (MLPs)[6]. Early approaches often relied on a single cellular feature set, such as mutation status or gene expression profile[7]. However, significant improvements have been achieved by incorporating multimodal information, such as chemical structure and pharmacological features[8,9]. These advancements have demonstrated the value of integrating diverse types of data to enhance drug response prediction.

Deep learning has become a way to effectively represent and integrate diverse feature sets. These methods commonly employ separate feature encoders that learn rich representations prior to integration[10,11]. For example, variational auto-encoders (VAEs) can leverage pretraining for transfer learning[12–15]. More broadly, neural networks are adaptable to novel inputs, such as graph representations for chemical structures[16–18], and their composability enables feature integration techniques like cross-attention[18,19].

Interpreting the computational structure of predictive models themselves can inform on the underlying biology of compound response. On one hand, ensemble models offer confidence scores, and direct interpretation of model coefficients (e.g., attention matrices) reveals feature relationships[19–22]. Gradient-based attribution methods can also help identify features driving a particular prediction[15]. On the other hand, integrating biological priors into neural networks effectively reduces the feature space and incorporates interpretable features, such as gene ontologies and pathway annotations[23–25]. By imposing constraints

[1]Department of Pharmaceutical Chemistry, University of California, San Francisco, San Francisco, CA, USA. [2]Institute for Neurodegenerative Diseases, University of California, San Francisco, San Francisco, CA, USA. [3]Bakar Computational Health Sciences Institute, University of California, San Francisco, San Francisco, CA, USA. [4]Department of Biochemistry and Biophysics, University of California, San Francisco, San Francisco, CA, USA. [5]Department of Urology, University of California, San Francisco, San Francisco, CA, USA. [6]Helen Diller Family Comprehensive Cancer Center, University of California, San Francisco, San Francisco, CA, USA. ✉e-mail: keiser@keiserlab.org

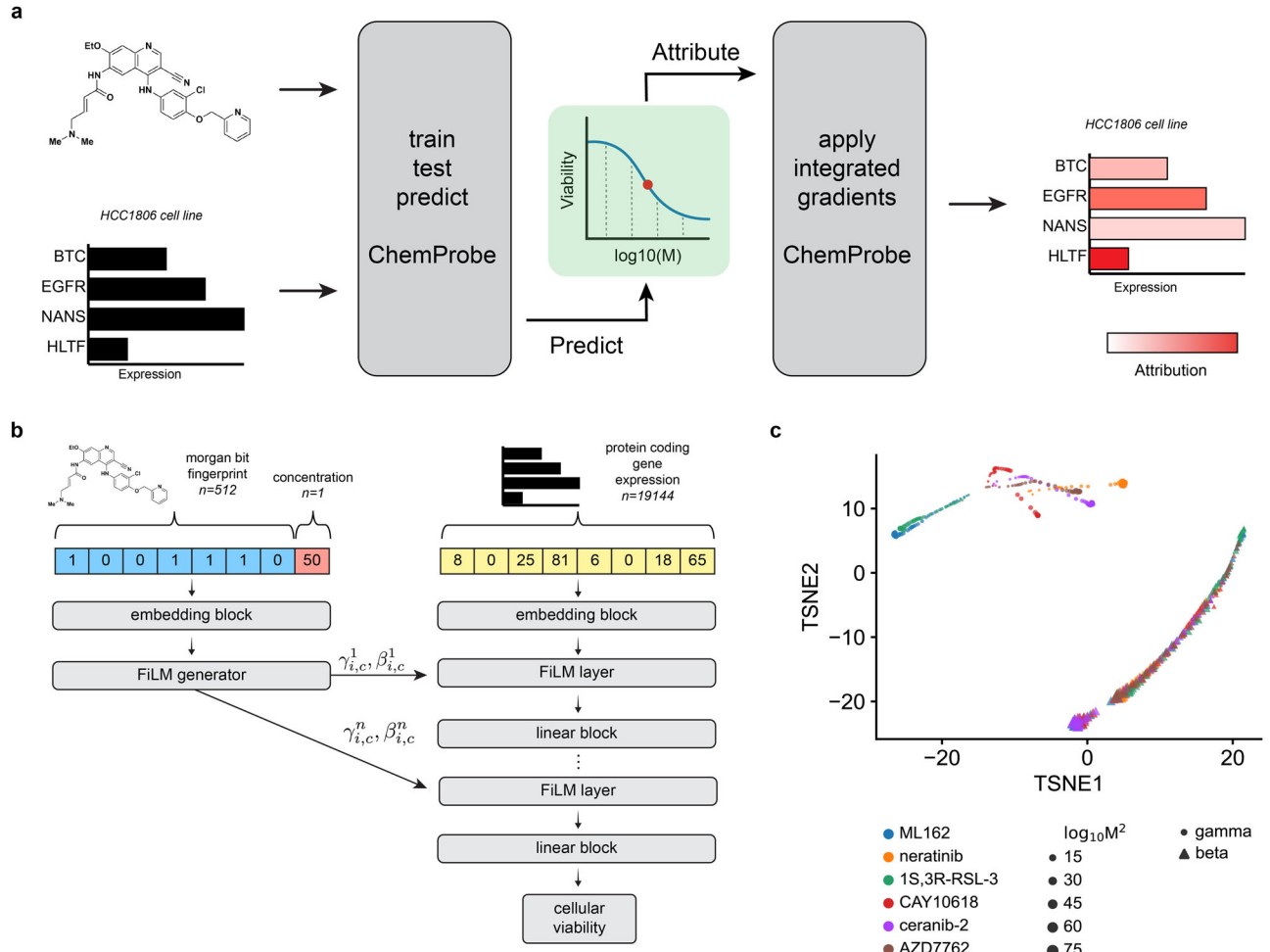

**Fig. 1 | ChemProbe design and model interpretation. a** Workflow of model training, validation, prediction, dose-response modeling, and feature attribution. We trained a deep neural network model to predict drug sensitivity at specific compound concentrations and fit log-logistic models to predictions. We derived compound pharmacodynamics from dose-response curves and applied integrated gradient saliency mapping to predicted IC50 to derive input feature attributions. IC50, inhibitory concentration of compound at 50% cellular viability. **b** Architecture of the conditional neural network (ChemProbe) trained to predict cell line viability from molecular features and compound structure. ChemProbe learns an embedding of protein-coding gene expression features conditioned by parameters learned from an embedding of compound structure and concentration. **c** Decomposition of learned conditioning parameters. Points represent compound-concentration samples; color indicates compound; size indicates concentration; and shape indicates parameter.

through priors, learning operates in the context of existing biological knowledge. However, this approach can limit a model's capacity to learn novel gene combinations and systems mechanisms that are not well understood.

We developed a conditional deep-learning model that predicts the sensitivity of new cellular samples to a panel of chemicals along with a framework for understanding the gene features implicated in the response. While previous research has addressed the tradeoffs among various input feature modalities, few studies explore how diverse feature sets are integrated. Moreover, little work has been done to explore if the features relied on by well-performing models indeed reflect expected biological relationships. Our work investigates methods for integrating biological and chemical features to improve drug sensitivity prediction and assesses the utility of model interpretation methods for advancing biological discovery. ChemProbe learns to combine cellular transcriptomes and chemical structures to predict sensitivity. The model can be applied to new biological samples and leverages integrated gradients to generate interpretable learned gene features relevant to known compound mechanisms. ChemProbe accurately models chemical response without biological priors, enabling in silico chemical screening of biological models and mechanistic interpretation of learned gene dependencies.

## Results

### Conditional modeling enhances cellular drug sensitivity prediction

We hypothesized that a deep neural network model could learn to combine gene expression with chemical structure to predict cellular sensitivity (Fig. 1a). We leveraged publicly available datasets to match cancer cell line basal transcriptomes with a large-scale chemical screen. The Cancer Therapeutics Response Portal (CTRP) reports the viability of 842 cancer cell lines in response to 545 compounds and compound pairs across a range of concentrations[26,27]. These compounds span cell circuitry targets, offering a nuanced view of cellular response to pathway perturbations across a broad range of cellular components. The Cancer Cell Line Encyclopedia (CCLE) provides basal transcriptomic characterizations of all 842 CTRP cell lines[28]. We combined compound structures and concentrations from the CTRP with protein-coding gene transcriptomes from the CCLE to create a dataset of compound-cell line pairs consisting of approximately 5.8 million labeled examples (Methods).

We formulated the cellular drug sensitivity prediction task as a conditional model $y = f(x|n)$, where $y$ is cellular viability, $x$ is a matrix of standardized RNA abundance values, $n$ is a matrix of chemical features, and $f$ is parameterized by a neural network (Methods). Thus the model's prediction of cellular viability depends on a cell's transcriptomic profile in the

## Table 1 | Predictive performance

| Model | $R^2$ |
|---|---|
| Concatenation | 0.6066 ± 0.0165 |
| Shift | 0.7060 ± 0.0304 |
| Scale | 0.7113 ± 0.0081 |
| FiLM | 0.7089 ± 0.0040 |
| Sructural ablation | 0.3016 ± 0.0304 |

Average performance and standard error of 5 models trained across identical data folds.

context of a chemical structure and concentration. ChemProbe predicts viability by learning to use chemical features to modulate gene expression through linear transformations of internal gene expression representations (Fig. 1b). This enables a logic akin to chemical substructures modulating gene products (proteins). We tested several ways to combine cellular features and chemical information within a single model, as assessed by the average maximum coefficient of determination ($R^2$). Accounting for comparable model sizes, we trained, validated, and hyperparameter-optimized different model architectures across five data folds stratified by cell line (five-fold cross-validation, Methods). We compared three methods of learned feature conditioning against a baseline feature concatenation approach[29]. All conditioning approaches outperformed feature concatenation by a notable margin (Table 1 and Supplementary Data 1). Among the conditioning models, scaling, shifting, and linearly modulating gene expression by chemical features performed similarly. A t-distributed stochastic neighbor embedding (t-SNE) decomposition of learned parameters demonstrated that scaling and shifting operations encoded distinct chemical features (Fig. 1c; Supplementary Data 2). Hierarchical clustering of scaling ($\gamma$) parameters grouped compounds by identity (Supplementary Fig. 1a and Supplementary Data 3), whereas compound concentration correlated with the first principal component of shifting ($\beta$) parameters ($p = 1.72e-55$; Supplementary Fig. 1b and Supplementary Data 3). Thus the learned conditioning parameters interpretably reflected compound structure and concentration in the drug-response modeling task as an emergent property of model learning.

Cellular response commonly follows a sigmoidal relationship to drug concentration. To quantify whether compound dosage alone was driving drug sensitivity predictions, we performed a feature ablation experiment, wherein we purposefully removed crucial data from the model's training and compared it to the actual model. For the "straw model,"[30,31] we replaced chemical fingerprints with unique but structurally uninformative and randomized numerical values. The "straw model" trained on ablated features failed, underscoring the importance of compound structural features in the modeling task (Table 1 and Supplementary Data 1)[31]. Explicitly modeling chemical information as conditioning provides a valuable inductive bias for chemical sensitivity prediction and gives insights into the predictive mechanisms of the model. We hyperparameter-optimized 5 FiLM models across cell-line stratified data folds and used this ensemble ($0.7173 \pm 0.0052$ $R^2$) of models in subsequent experiments (Methods).

### ChemProbe predicts breast cancer patient response

We next asked whether learned transcriptional patterns would generalize to an in vivo cellular context. We measured how well ChemProbe, trained solely on cell line expression profiles, could predict drug response in clinical tumor samples. We used gene expression and patient-drug response data from the I-SPY2 adaptive, randomized, phase II clinical trial of neoadjuvant therapies for early-stage breast cancer (NCT01042379)[32,33]. I-SPY2 assigned patients to treatment arms based on biomarkers such as hormone receptor status, human epidermal growth factor receptor-2 expression, and MammaPrint status. The absence of invasive cancer in the breast and regional lymph nodes at the time of surgery defined the endpoint of pathological complete response (pCR) (nonresponse, pCR = 0; response, pCR = 1).

The I-SPY2 dataset introduced a significant change in the input data modality and allowed us to assess the robustness of ChemProbe. Unlike the CCLE training data, which quantified gene expression through high-throughput RNA sequencing, I-SPY2 collected pre-treatment patient gene expression by microarray. Microarrays have lower overall specificity and sensitivity and capture a smaller dynamic range of gene abundance[34]. Neural networks often fail on "out of distribution" samples whose features (gene abundance values) come from different assays than their training data. To determine the extent to which the I-SPY2 data was outside ChemProbe's training distribution, we compared dataset expression profiles across the top two principal components. A subset of the I-SPY2 data fell outside the training data distribution, consistent with the expectation that assay types introduce systematic measurement effects (Methods, Supplementary Fig. 2a and Supplementary Data 4).

We assessed whether ChemProbe could retrospectively stratify I-SPY2 responders and non-responders despite these differences in datasets. We compared ChemProbe predictions with the original treatment allocations in the I-SPY2 trial, which were determined by standard biomarkers of the participants' tumors. Five drugs from the I-SPY2 trial were in ChemProbe's panel. We first compared the magnitudes of ChemProbe's sensitivity predictions between responders and non-responders. For four out of five drugs, ChemProbe predicted lower scaled-AUC values for the responder group (Fig. 2a and Supplementary Data 5). Next, we generated receiver operating characteristic (ROC) curves to compare the drug response predictions of I-SPY2 and ChemProbe with the trial outcomes. We used the treatment designations from I-SPY2 as a proxy for drug response prediction. ChemProbe's area under the ROC curve for each drug ranged from 0.60 (paclitaxel and neratinib) to 0.73 (veliparib), with a macro-average auROC of 0.65 (Fig. 2b and Supplementary Data 5).

To evaluate the clinical utility of ChemProbe, we used it to classify patients by treatment response: responders (+) and non-responders (−) (ChemProbe + /−). Since the model determines cellular viability by drug concentration, we established a decision threshold for detecting responders (Methods). ChemProbe + /− classification accuracy significantly outperformed I-SPY2 ($p < 5e-2$; Fig. 2c and Supplementary Data 5). Although I-SPY2 predictions had a higher true positive rate (0.30, I-SPY2; 0.21, ChemProbe), ChemProbe + /− classifications massively reduced the false positive rate (0.70, I-SPY2; 0.37, ChemProbe) with relatively few false negatives (0.00, I-SPY2; 0.095, ChemProbe) (Supplementary Fig. 2b, c and Supplementary Data 4). By correctly predicting a portion of patients with a low likelihood of drug response, ChemProbe + /− significantly increased the true negative rate of drug-response classification relative to I-SPY2, providing crucial information for clinical decision-making. Despite being trained only on isogenic cell lines, these results support ChemProbe's use with heterogeneous tumors from clinical patient samples.

### ChemProbe predicts cellular drug sensitivity

We conducted a prospective evaluation of ChemProbe's ability to differentiate drug sensitivity between two primary breast cancer cell lines, HCC1806-Par and MDA-MB-231-Par[35–38]. We compared the gene expression profiles of the two cell lines to their CCLE counterparts by analyzing the top two gene-expression principal components. Our analysis showed significant disparities in the gene expression patterns of the two cell lines compared to the training data, highlighting the challenges of maintaining consistency across cellular models (Supplementary Fig. 3a and Supplementary Data 6). The observed differences, which make the prospective test more difficult but particularly informative, may be attributed to variations in cell culture protocols, reagents, and genetic drift commonly found between experimental settings[39].

We predicted sensitivity at 32 drug concentrations (1e-3 μM–300 μM), fit log-logistic models, and determined 50% inhibitory concentration (IC50) values from each in silico dose-response curve (Fig. 3a). ChemProbe predicted that HCC1806-Par would be more sensitive than MDA-MB-231-Par to 88.16% (201/228) of the compounds with fitted curves (Fig. 3b and Supplementary Data 7). We focused on compounds with the largest

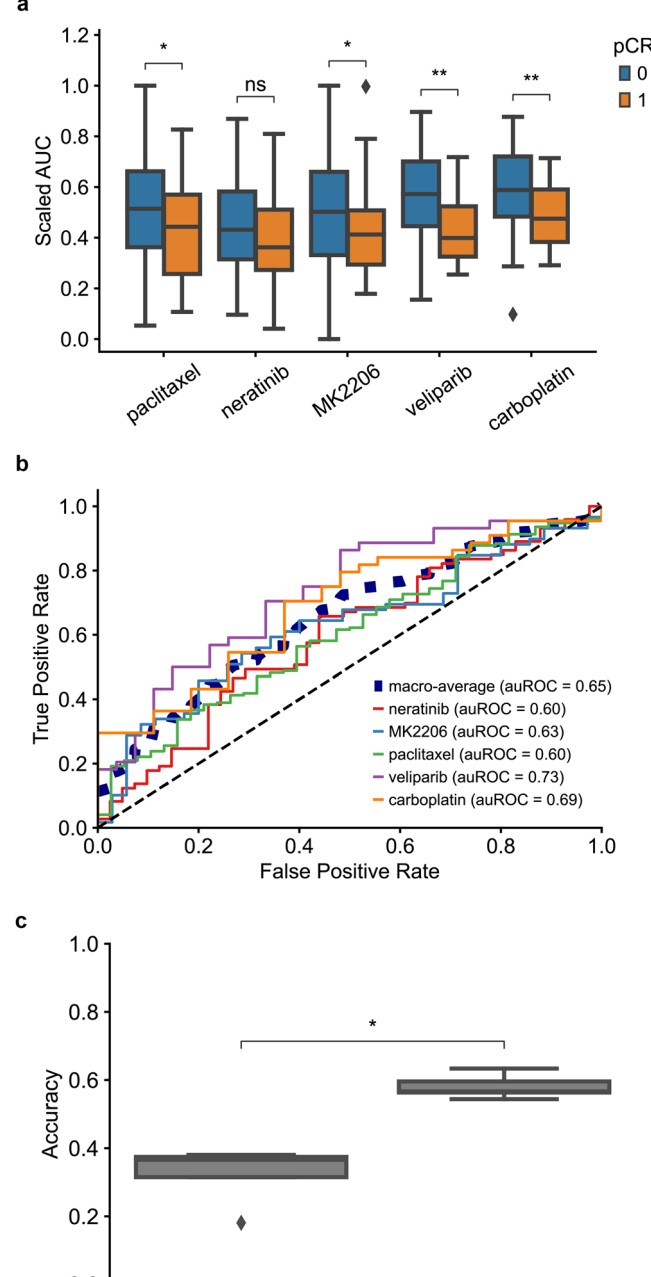

**Fig. 2 | I-SPY2 clinical trial retrospective analysis. a** Predicted dose-response AUC for I-SPY2 patients treated with each drug. AUCs scaled between the minimum and maximum predicted AUC of patients treated with each drug. Blue = non-responder, orange = responder; centerline, median; box limits, upper and lower quartiles; whiskers, 1.5x interquartile range; points, outliers; two-sided Wilcoxon rank-sum test; ns: $p < =1e1$, $*p < =5e-2$, $**p < =1e-2$. Sample sizes: paclitaxel ($n = 38$ independent case samples, $n = 172$ independent control samples); neratinib ($n = 41$ independent case samples, $n = 73$ independent control samples); MK2206 ($n = 35$ independent case samples, $n = 59$ independent control samples); veliparib ($n = 27$ independent case samples, $n = 44$ independent control samples); carboplatin ($n = 27$ independent case samples, $n = 44$ independent control samples). **b** Receiver operating characteristic curve of patients treated with each drug and corresponding auROC. **c** Accuracy of I-SPY2 ($n = 5$ independent samples) predictions versus ChemProbe ($n = 5$ independent samples) predictions for non-responders/responders. Centerline, median; box limits, upper and lower quartiles; whiskers, 1.5x interquartile range; points, outliers); two-sided Wilcoxon rank-sum test; $*p < =5e-2$.

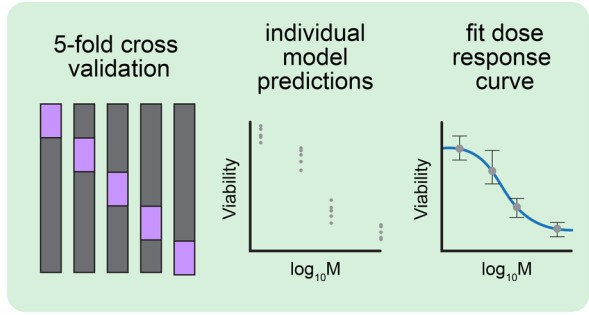

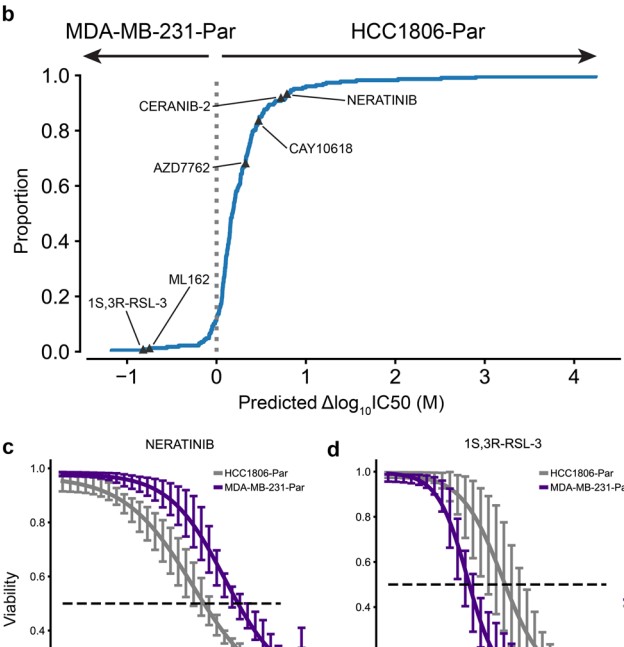

**Fig. 3 | Differential potency and in silico dose-response curve predictions. a** Approach to model training and dose-response modeling. We trained individual models on held-out cell line dataset splits by 5-fold cross-validation. We then fit log-logistic models to cross-validated model predictions and derived pharmacodynamic features. **b** Expected cumulative distribution plot of predicted compound IC50 differences between HCC1806-Par and MDA-MB-231-Par cell lines. Compounds selected for in vitro dose-response testing are highlighted. **c** Predicted dose-response relationships of HCC1806-Par and MDA-MB-231-Par response to neratinib ($n = 5$ independent samples) and **d** 1S,3R-RSL-3 ($n = 5$ independent samples). 95% confidence intervals.

differences in IC50 between the cell lines, selecting four compounds predicted to have strong IC50s against HCC1806-Par (neratinib, ceranib-2, CAY10618, and AZD7762) and two compounds with IC50s favoring MDA-MB-231-Par (ML162 and 1S,3R-RSL-3) (Fig. 3c, d; Supplementary Fig. 4 and Supplementary Data 7). In vitro, prospective testing confirmed ChemProbe's predictions for all six compounds. Predicted differences in IC50s between the two cell lines significantly correlated with observed differences (Fig. 4a–f, $p = 0.035$; Fig. 4g, Supplementary Data 8). We compared the relative potency of each compound at the median effective dose (ED50) between cell lines, finding significant differences in compound cellular viability as predicted (Table 2). Additionally, predicted IC50s correlated highly with measured IC50s for individual cell lines after correcting for an

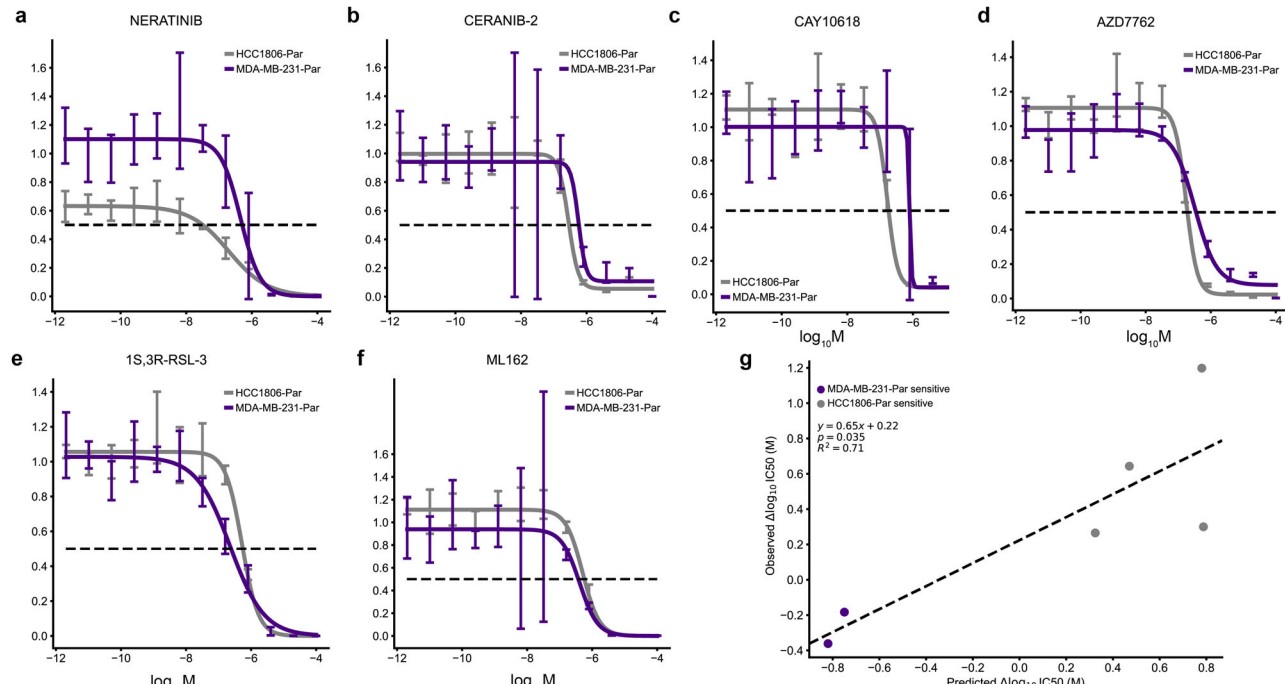

**Fig. 4 | Validation of differential potency predictions. a–d** In vitro dose-response relationships of HCC1806-Par differentially potent compounds (*n* = 4 independent experiments) and **e, f** MDA-MB-231-Par differentially potent compounds (*n* = 4 independent experiments). 95% confidence intervals. **g** Relationship between the predicted and observed differences in IC50 values of tested compounds between HCC1806-Par and MDA-MB-231-Par cell lines. Two-sided *t*-test (*n* = 6 independent experiments).

experimental outlier (*p* = 0.096, Supplementary Fig. 5a and Supplementary Data 9). These results were consistent with initial concentration range-finding experiments, where five out of six compounds had shown predicted differences in IC50s and relevant IC50s specific to individual lines (Supplementary Fig. 5b, c and Supplementary Data 9). ChemProbe accurately predicted the sensitivities of independently obtained and characterized cell line samples, despite their transcriptomic profiles differing from the training dataset.

**Gene expression attribution vectors pass interpretability soundness checks**

To assess whether ChemProbe learned biologically relevant patterns, we investigated whether the model's gene expression saliency reflected known compound pharmacology and network biology. Saliency mapping methods evaluate which gene features a model puts the most "weight" on when making its predictions and how these choices change for different cell lines or compounds. We experimentally characterized seven new cell line transcriptomes, including primary tissue models and metastatic derivatives[40–42]. Using integrated gradient saliency mapping[43], we determined IC50 values for each compound-cell line pair with ChemProbe and computed its gene attribution vectors. However, we acknowledge that integrated gradient attribution vectors may correlate with input feature magnitudes, potentially

undermining their usefulness in quantifying feature importances[44]. We used principal component analysis to determine the first two principal components of attribution vectors by cell line to check this. We found that attribution and transcriptome vectors correlated (Supplementary Fig. 6a, c and Supplementary Data 10). Although some gene expression magnitudes may linearly correlate with phenotypes, this framework does not capture other causal nonlinear interactions within gene regulatory networks. A well-calibrated interpretation method should attribute significance to features based on their relevance to the model's predictions, rather than solely on their expression levels. To address this confounder, we normalized attribution vectors by cell line, which decreased cell-line-specific effects in the principal component analysis and decoupled the correlation between attribution and transcriptome vectors (Methods, Supplementary Fig. 6b, c and Supplementary Data 10).

To ensure model interpretation accurately reflected learned feature transformations, we performed two tests (Methods)[44]. The first test randomly initialized the model parameters and compared the outcomes with the true-model's attribution vectors. We trained the model using scrambled labels in the second test and compared its attribution vectors with the true-model's. We conducted these tests using both uncorrected (raw) and cell line-effect corrected (adjusted) attribution vectors. The raw attribution vectors were highly correlated with transcriptome profiles, random-model, and permuted-model attribution vectors, failing the tests of independence from learned parameters and the training data. However, the adjusted attribution vectors were not correlated with those derived from the control models, indicating that adjusted attribution vectors do not simply reflect data or architecture artifacts (Supplementary Fig. 6c and Supplementary Data 10).

**Learned transcriptomic features reflect compound pharmacology and network biology**

Neural networks are notoriously difficult to interpret, but we hypothesized that ChemProbe's highly attributed gene features may reflect causative mechanisms or correlative biomarkers of drug sensitivity. First, we investigated whether the model relied on similar gene features for compounds

## Table 2 | Relative potency at median effective dose

| Compound | ED50 ratio (HCC1806-Par/MDA-MB-231-Par) | *t*-value | *p*-value |
|---|---|---|---|
| Neratinib | 0.4946 ± 0.2426 | −2.0830 | 4.0220e-2 |
| Ceranib-2 | 0.5165 ± 0.1943 | −2.4887 | 1.4700e-2 |
| CAY10618 | 0.2089 ± 1.869e-2 | −42.3233 | 1.1593e-59 |
| AZD7762 | 0.5639 ± 0.1004 | −4.3430 | 3.7500e-5 |
| 1S,3R-RSL-3 | 2.1123 ± 0.4446 | 2.5244 | 1.3384e-2 |
| ML162 | 3.008 ± 0.7041 | 2.8521 | 5.4120e-3 |

Relative potency of each compound at the median effective dose (ED50) between cell lines.

with the same known protein targets. We created a control compound set (CCS) based on nominal target classes, each with at least two compounds successfully predicted in all seven cell lines. We applied *K*-means clustering to the CCS attribution vectors and computed the adjusted mutual information (AMI) between clusters and target class labels to determine whether transcriptomic attribution vector similarity corresponded to known compound mechanisms of action (MOA) (Fig. 5a and Supplementary Data 11). We also examined the AMI between structural clusters and target classes, as chemical structure similarity alone may at times reflect target profile similarity, albeit imperfectly[45].

The attribution vector clusters AMI was significantly greater than that of structural clusters, a randomly initialized model, and a model trained on permuted labels (Fig. 5b, Methods; Supplementary Data 11). Moreover, we found that compounds belonging to the same target class frequently had high nominal target attributions relative to other compounds, indicating that ChemProbe often made predictions based on the expression information of nominal targets (Fig. 5c; Supplementary Fig. 6d–k and Supplementary Data 10–11).

We next examined the network topology of nominal target classes using the STRING database of high-confidence protein–protein interactions[46] to interrogate biological relevance. We clustered attribution vectors, gathered target annotations within each cluster, and queried STRING for the respective target interactome (Fig. 5d, Methods; Supplementary Data 11). Target modules had significantly greater connectivity than modules generated from randomly sampled target protein sets or randomly sampled protein sets (Fig. 5e; Supplementary Data 11). Finally, we tested whether attribution-defined target modules of action (ModOA) also showed protein interaction enrichment. On analysis, 10/26 ModOA reflected significant network interaction enrichment and a variety of functional enrichments from gene ontologies, KEGG pathways, and Reactome pathways (Fig. 5f; Supplementary Fig. 7 and Supplementary Data 11, 12). These findings suggest that highly attributed transcriptome features reflect systems biology and potential mechanisms of drug response.

### Screening genetic dependencies for mechanisms of ferroptosis

We further hypothesized that ChemProbe's highly attributed gene features would relate to compound MOA. To test this, we used linear regression for differences in gene attribution between groups. This "differential attribution analysis" (DAA; see Methods) generates ranked gene lists, which we use as marker genes to arrange attribution clusters hierarchically (Fig. 6a and Supplementary Data 13). We noticed clusters 26 and 28 showed different prediction sensitivity to ferroptosis-inducing compounds (Fig. 6b and Supplementary Data 13). Ferroptosis is a type of cell death implicated in multiple biological contexts, with therapeutic applications in cancer, immunity, development, and aging[47,48]. These attribution clusters included compounds ML162 and 1S,3R-RSL-3, which had shown differential cellular sensitivity in the prospective in vitro experiments (Fig. 4e, f). Additional compounds with ferroptosis-inducing mechanisms of action in these clusters included ML210, erastin, CIL56, and CIL70.

Next, we investigated the attribution vectors of ferroptosis-inducing compounds to assess the alignment of model interpretations with established ferroptosis biology. We observed a clear distinction in predicted sensitivity between two groups of cell lines exposed to the same compounds (Fig. 6b and Supplementary Data 13). To further analyze this, we merged clusters 26 and 28 into a combined cluster representing ferroptosis-inducing compounds and applied DAA. Since multiple mechanisms induce ferroptosis, we queried differential attributions of multiple ferroptosis-associated genes, including *GPX4*, *SCD*, *SLC7A11*, *FSP1*, and *LRP8*[47]. All ferroptosis-associated genes were within the most highly attributed in the ferroptosis-inducing compound cluster (Fig. 6c and Supplementary Data 13). To verify that these results were not artifacts of the transcriptomes or relative gene expression differences, we also performed differential expression analysis (DEA) between MDA-MB-231-Par and HCC1806-Par. Besides *GPX4*, a key ferroptosis regulator, no ferroptosis-associated genes rose to significance ($p < 5e-2$; Supplementary Fig. 8a and Supplementary Data 14).

Changes in compound sensitivity following gene knockout (KO) or overexpression can inform on mechanisms of gene-dependent protection or resistance. Accordingly, we assessed ChemProbe's utility for screening gene-dependent ferroptosis resistance in silico. Lipoprotein receptor *LRP8* has recently been shown to act as a ferroptosis resistance factor by maintaining cellular selenium levels and appropriate translation of *GPX4*. Selenium uptake is reduced in LRP8 KO models, leading to ribosome stalling and early translation termination of *GPX4*, which sensitizes cells to ferroptosis[49]. We tested if ChemProbe correctly predicted that an LRP8 KO cell line would have reduced sensitivity to ferroptosis-inducing compounds than a wild-type. Consistent with previous research, ChemProbe predicted LRP8 KO cells were more sensitive than wild-type to known ferroptosis-inducing compounds ML210, 1S,3R-RSL-3, ML162, and CIL56 (Fig. 6d and Supplementary Data 13).

We noticed several correlations between cellular response and the expression of highly attributed genes for compounds that induce ferroptosis (Supplementary Fig. 8b, c). We wondered if highly attributed genes played functional roles related to ferroptosis. We extracted the ten highest differentially attributed genes and applied a functional enrichment analysis (Supplementary Fig. 8d and Supplementary Data 14). We observed the enrichment of terms related to lipid transport and fatty acid metabolic processes, pathways adjacent to lipid peroxidation, and ferroptosis (Fig. 6e and Supplementary Data 13). These results indicate that transcriptomic attributions align with ferroptosis biology, underscoring the potential of ChemProbe in screening genetic dependencies and identifying novel biological mechanisms.

## Discussion

Using a conditional deep-learning approach, ChemProbe evaluates cellular transcriptomic signatures against bioactive molecular structures to predict cellular responses to chemical perturbations. In experiments on cellular models and clinical tumor samples, this tool accurately predicts cellular viabilities. ChemProbe complements more clinically oriented approaches with its ability to directly screen engineered cell lines and interrogate potential molecular mechanisms. Engineered cell lines, which possess specific genetic modifications or alterations, physically model disease conditions or the results of targeting pathways of interest. By leveraging ChemProbe, researchers can evaluate the sensitivity of known and newly engineered lines to a panel of chemical probes to assess how specific genetic modifications or alterations influence compound response.

Intriguingly, deep-learning model interpretation reflects compound mechanisms of action (Fig. 5). The differential attribution analysis (DAA) method we introduce surfaces potential gene patterns driving responses and new disease-gene relationships. In one example, we identified genes linked to ferroptosis resistance in an *LRP8* knockout cell line. ChemProbe's calculations were not specific to this biology; it may find similar use in screens for resistance mechanisms and target discovery across diverse cellular models (Fig. 6). In cancer research, the tool rapidly evaluates the influence of specific oncogenic mutations or alterations in tumor suppressor genes on chemical sensitivity. When applied to engineered cell lines representing different genetic backgrounds, ChemProbe can highlight vulnerabilities and potential mechanisms of drug resistance associated with particular genetic alterations.

Extending to clinical samples, ChemProbe becomes a tool for targeted therapy and precision medicine. We found that it predicts drug sensitivity in breast cancer patients across heterogeneous clinical tumor samples (Fig. 2). Likewise, ChemProbe suggests which drugs may be ineffective for a given patient; if borne out in clinical studies, it or similar methods could meaningfully reduce therapeutic trial and error[50,51]. Although our data may suggest alternative drug treatments, we refrained from recommending therapies as their potential value is challenging to evaluate within the scope of this study. The ability to expedite treatment at earlier disease stages and target cellular vulnerabilities would be particularly impactful for tumors whose resistance mechanisms rapidly evolve.

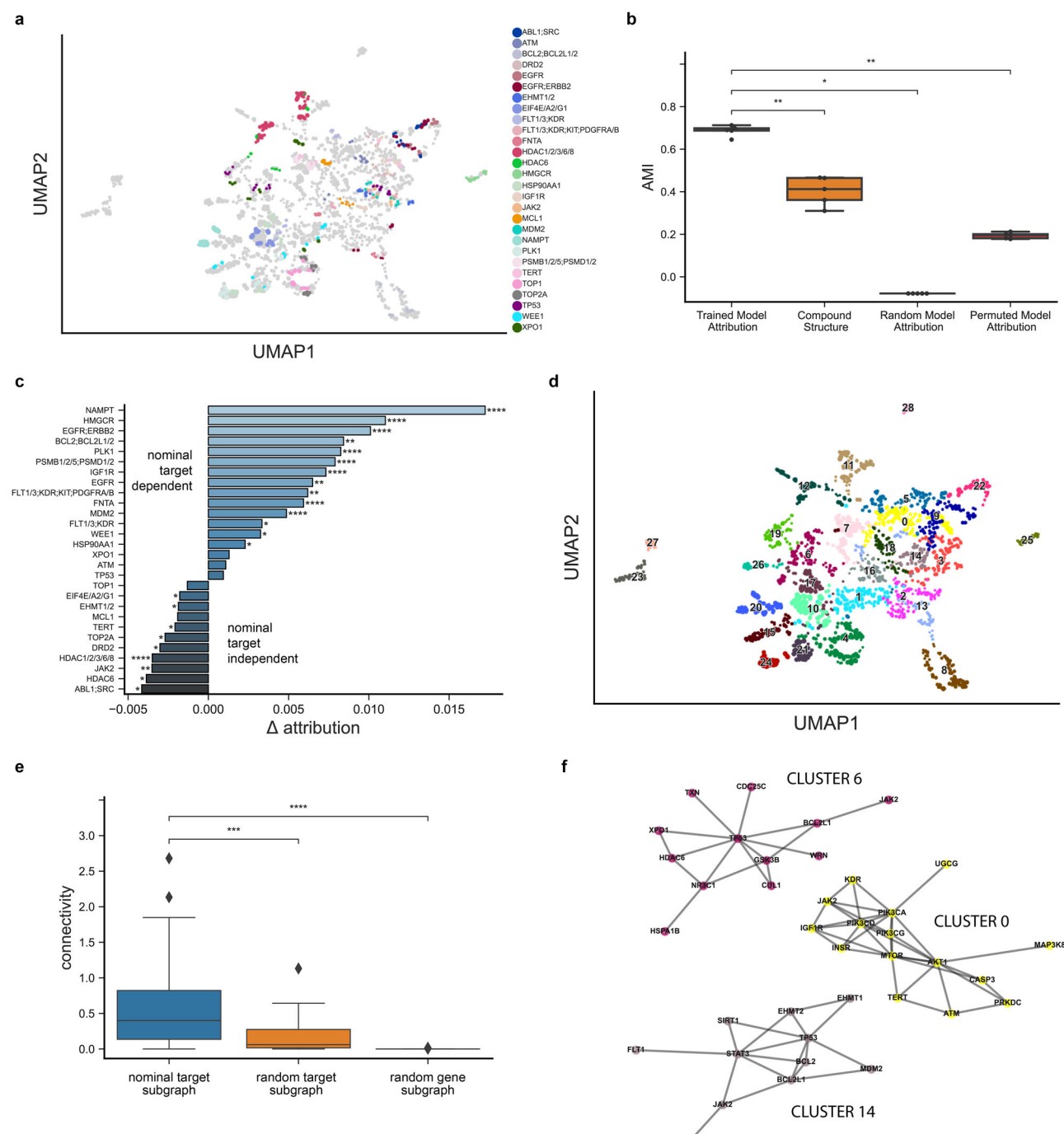

**Fig. 5 | Feature attribution analysis of nominal compound targets. a** Distinct protein target clusters emerge from UMAP decomposition of adjusted attribution vectors at compound IC50s for predicted and fitted dose-response relationships in MDAMB231, MDAMB231-LM2, HCC1806, HCC1806-LM2b/c, SW480, and SW480-LvM2 prospective cell lines. Control compound set (CCS) attribution vectors colored by nominal target class. **b** Comparison of adjusted mutual information (AMI) derived from CCS nominal target labels and *K*-means clustering of trained model adjusted attribution vectors (*n* = 5 independent samples), compound fingerprints (*n* = 5 independent samples), random-model adjusted attribution vectors (*n* = 5 independent samples) and permuted-model adjusted attribution vectors (*n* = 5 independent samples). Centerline, median; box limits, upper and lower quartiles; whiskers, 1.5x interquartile range; points, outliers; two-sided Wilcoxon rank-sum test; *\*p* < =5e-2, *\*\*p* < =1e-2. **c** Average attribution difference between the

highest significance target of the nominal target class versus all other target classes. Two-sided Wilcoxon rank-sum test; *\*p* < =5e-2, *\*\*p* < =1e-2, *\*\*\*p* < =1e-3, *\*\*\*\*p* < =1e-4. Sample sizes: *n* = 14 independent case samples, *n* = 455 independent control samples except otherwise noted; EGFR;ERBB2 (*n* = 21 case, *n* = 448 control); HDAC1/2/3/6/8 (*n* = 56 case, *n* = 413 control); HMGCR (*n* = 21 case, *n* = 448 control); NAMPT (*n* = 28 case, *n* = 441 control); TP53 (*n* = 21 case, *n* = 448 control). **d** Leiden clustering of all attribution vectors. **e** Comparison of PPI subgraph connectivity derived from clustered target profiles (*n* = 26 independent samples), random target profiles (*n* = 26 independent samples), and random protein-coding genes (*n* = 26 independent samples). Centerline, median; box limits, upper and lower quartiles; whiskers, 1.5x interquartile range; points, outliers; two-sided Wilcoxon rank-sum test; *\*\*\*p* < =1e-3, *\*\*\*\*p* < =1e-4. **f** Network representation of exemplar clustered target profile subgraphs.

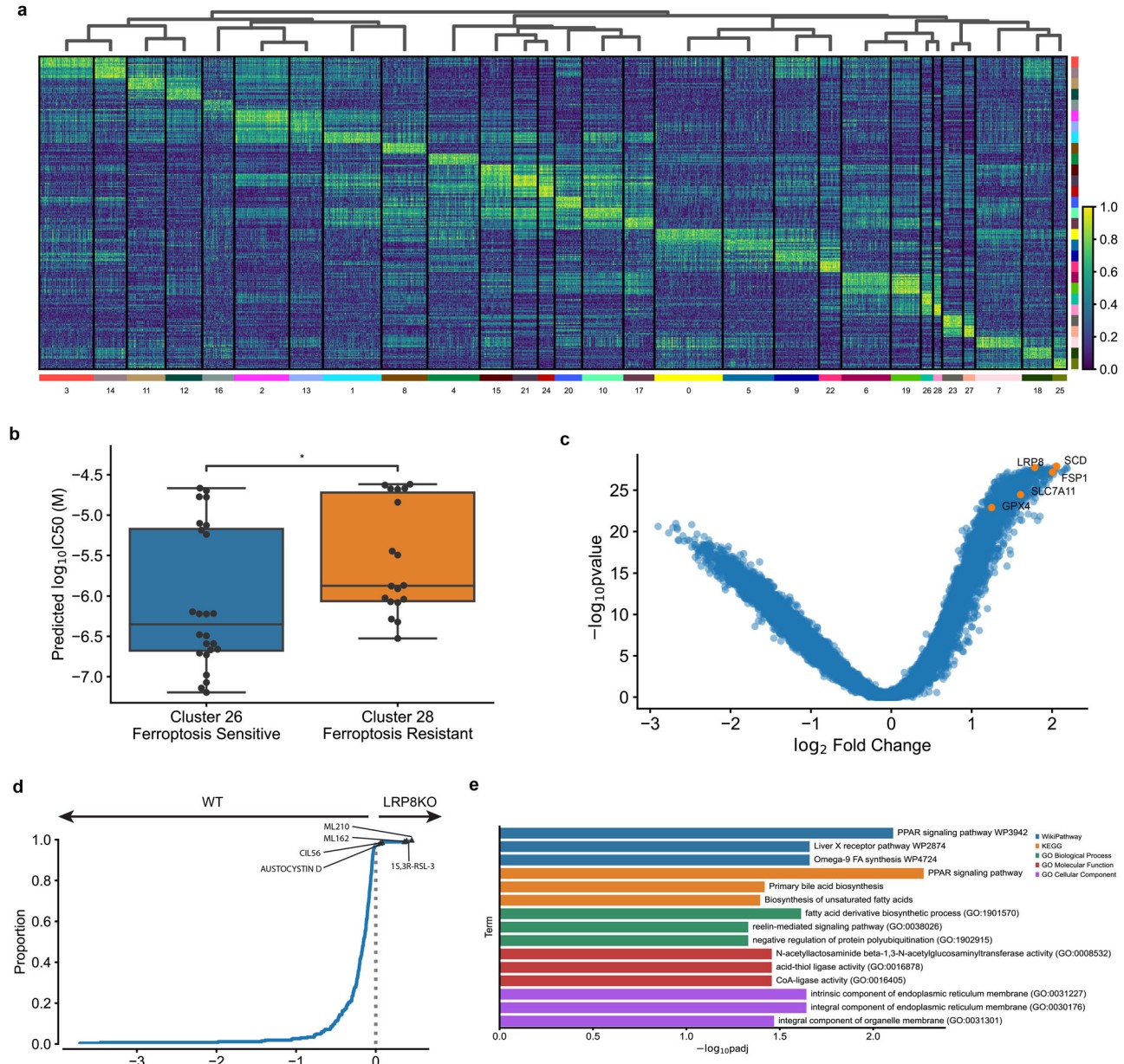

**Fig. 6 | Differential attribution analysis (DAA) of ferroptosis-inducing compounds. a** Heatmap of top-10 differentially attributed genes within Leiden clusters from Fig. 5d. Clusters ordered by hierarchical clustering of DAA profiles (columns). Rows: top-10 attributed genes, columns: cell line-compound attribution sample. **b** Comparison of predicted IC50s between cluster 26 (ferroptosis-sensitive, $n = 24$ independent samples) and cluster 28 (ferroptosis-resistant, $n = 18$ independent samples). Centerline, median; box limits, upper and lower quartiles; whiskers, 1.5x interquartile range; points, outliers; two-sided Wilcoxon rank-sum test; *$p < =5e-2$.

**c** Volcano plot of DAA results derived by comparing ferroptosis-inducing compound attributions to all other compound attributions. Known ferroptosis-mediating genes in orange. **d** Expected cumulative distribution plot of predicted compound IC50 differences between HCC1143 WT and LRP8 KO cell lines. Ferroptosis-inducing compounds predicted differentially potent in LRP8 KO marked. **e** Enrichment analysis of top-10 differentially attributed genes of ferroptosis-inducing compound samples. Fisher's exact test.

Nonetheless, several caveats merit mention. The experiments were constrained to a limited set of cell lines and compounds, and model interpretation reflects a limited set of biological factors. Without further study, gene features achieving high model attribution may reflect useful but obtuse patterns in the datasets rather than biological causality. Similarly, without a more diverse training set of chemical structures, the model may not be leveraging generalizable structural features. Deep learning model attribution methods are primarily empirical, so the potential compound mechanisms of action they reveal ultimately necessitate prospective biological testing[52,53].

ChemProbe screens cell lines against half a thousand chemical probes and drug-like compounds. However, expanding its predictions to a larger

subset of chemical space would require collecting biological screening training data at a commensurate scale, which is currently impractical. Deep learning models like AlphaFold and ESM have leveraged self-supervised learning to extract emergent properties from extensive unlabeled protein sequence data[54,55]. Similarly, integrating ChemProbe with pre-trained cellular transcriptomic or small-molecule structure foundation models may be a means to expand into broader biological and chemical space.

When used to screen disease models, engineered cell lines, and clinical samples, ChemProbe is a powerful tool to assess how cells respond to various compounds. It supports exploring new therapeutic targets, suggests disease mechanisms, and can help researchers develop more precise and

effective treatments. ChemProbe's conditional architecture enables expressive sensitivity predictions across chemical concentrations, enhancing model interpretability and adaptability to new probes. In this study, ChemProbe predicted drug sensitivities in multiple contexts, from cancer cells and tumor samples to data on breast cancer patient responses. Looking forward, we hope that ChemProbe's availability as an open-source tool will contribute to a range of research efforts in precision medicine and beyond.

## Methods

### Pharmacogenomic dataset

Drug sensitivity data was obtained from the Cancer Therapeutic Response Portal v1 and v2 (CTRP v1/2). These datasets comprise 864 cell line responses to 481 individual compounds and 64 compound pairs across a range of concentrations. Response phenotypes were quantified by cellular viability, a normalized measure characterizing complete cell killing to cell stasis (0–1) and cell growth (>1). We utilized predicted cellular viability derived from fitted dose-response curves of each experimental set, in which replicate cell line-compound experiments were fit with a log-logistic function, and predicted cellular viability was derived at the original experimental concentrations[27]. The compound structure was represented as 512-bit Morgan fingerprints (radius = 2) converted by RDKit from SMILES provided by the CTRP. Experimental micromolar compound concentrations were concatenated with Morgan fingerprints, resulting in 513-length compound feature vectors. We matched CTRP cell lines with the Cancer Cell Line Encyclopedia (CCLE) molecular characterizations and extracted protein-coding gene expression measurements, resulting in 19,144-length cell line feature vectors. In total, 545 total compounds or compound pairs and 860 cell lines comprised 366,710 unique pairs and 5,849,340 total individual examples of compound response at various concentrations.

### ChemProbe architecture, training, and evaluation

The study focused on predicting drug sensitivity in the context of pharmacological intervention by integrating cell state features with compound features. To achieve this, we formulated a conditional model where cellular viability is predicted based on a vector of standardized protein-coding RNA abundance values and a vector of chemical features, including structure and concentration. We explored two methods of integrating gene expression and small-molecule feature representations: simple concatenation and hierarchical integration using feature-wise linear modulation (FiLM).

ChemProbe includes a conditional encoder that embeds compound features into a vector of length $c$ and an inputs encoder that embeds gene expression features into a vector of length $g$. We used a FiLM generator to predict $\gamma$ and $\beta$ parameters of length $g$ based on compound embeddings. The FiLM layer then applies an affine transformation of gene expression embeddings by $\gamma$ and $\beta$ parameters. This process repeats across $n$ FiLM layers, and the modulated gene expression embeddings pass through a linear block consisting of a linear layer, ReLU activation, batch normalization, and dropout. The final linear block compresses feature maps to a vector of length 1, and the mean-squared error is calculated between predicted cellular viability and true cellular viability. In our experiments and publicly available trained models, we use a transcriptome embedding block with layers of size [2048, 512, 256] and a compound embedding block with layers of size [256, 128] to project to an embedding of size $g = c = 128$. We used $n = 2$ FiLM layers in the final models.

To evaluate the performance of our model, we used cross-validation and split cell line-compound pairs into five groups of approximately equal size by cell line to avoid data leakage and performance inflation. We trained five individual models in a leave-one-out cross-validation scenario and applied 20 rounds of hyperparameter optimization to all five individually trained models. We implemented the ChemProbe model in PyTorch and applied hyperparameter optimization with Optuna.

### Dose-response assay and cell culture

MDA-MB-231-Par and HCC1806-Par cells were seeded at 1,000 cells per well in quadruplicate per condition in a white opaque 96-well plate (catalog no. 3917, Corning). Twenty-four hours later, cells were treated with serial dilutions between 2.05 pM and 100 μM of the following compounds: neratinib (catalog no. 18404, Cayman Chemical), CAY10618, 1S,3R-RSL-3 (catalog no. 19288, Cayman Chemical), AZD7762 (catalog no. 11491, Cayman Chemical), ceranib-2 (catalog no. 11092, Cayman Chemical), and ML162 (catalog no. 20455, Cayman Chemical), and DMSO control. Cells were treated for 72 h with media replaced every 24 hours. Cell viability was measured with the CellTiter-Glo 2.0 Assay (catalog no. G9243, Promega Corporation) with 1000 ms integration time.

All cells were cultured at 37 °C in a humidified incubator with 5% CO2. MDA-MB-231-Par (ATCC HTB-26) cells were grown in DMEM supplemented with 10% FCS, penicillin (100 U ml$^{-1}$), streptomycin (100 μg ml$^{-1}$) and amphotericin (1 μg ml$^{-1}$). HCC1806-Par (ATCC CRL-2335) cells were grown in Roswell Park Memorial Institute-1640 medium supplemented with 10% FCS, L-glutamine (2 mM), sodium pyruvate (1 mM), penicillin (100 U ml$^{-1}$), streptomycin (100 μg ml$^{-1}$) and amphotericin (1 μg ml$^{-1}$).

### Statistics and reproducibility

Details regarding sample sizes, number of replicates, and statistical methods are provided in the respective section subheadings.

**Predictive modeling baselines.** We compared different models that modify gene expression features by compound structure and concentration using various transformations. Our baseline model, "concatenation" architecture, simply combined gene expression and compound features into a single vector, which was fed into a multi-layer perceptron. We independently evaluated the isolated effects of learning transformation types using the "scale" and "shift" variants of the ChemProbe model. The "scale" model held shift parameters constant ($\beta=0$) and learned only the scale parameters ($\gamma$), whereas the "shift" model held scale parameters constant ($\gamma=1$) and learned only the shift parameters ($\beta$). We assessed ChemProbe's dependence on compound concentration by creating a "permuted" model that used random binary fingerprints for each compound, ablating structural information. We trained and evaluated all models using 5-fold cross-validation on the originally defined dataset splits for three rounds of hyperparameter optimization.

**Dose-response modeling.** To generate predicted dose-response curves, log-logistic functions were fit to each set of cell line-compound predictions obtained from the five individually trained ChemProbe models. A sequence of quality control conditions was defined to ensure the reliability of each dose-response relationship. Firstly, cellular viability at any of the four largest compound concentrations was checked for increases of 20% or more from the fifth largest compound concentration. If this condition was met, the viability prediction at the largest concentration was dropped. This process was repeated recursively, and a minimum of 16 data points was required for fitting a dose-response curve. If the minimum predicted cellular viability was greater than 0.4, no dose-response curve was fit. For cell line-compound pairs that passed quality control, a 4-parameter log-logistic function was fit. If the optimization failed, a 3-parameter log-logistic function was fit. If this optimization also failed, a 2-parameter log-logistic function was fit. Additional quality control was performed during the analysis of predicted dose-response curves by filtering out log-logistic functions with undetermined parameters and with predicted EC50 < 1e-3 or EC50 > 300. Scipy was used to fit parameters of log-logistic functions to dose-response relationships.

For relative potency comparisons, the drc package in R was employed to fit dose-response models with a four-parameter log-logistic model. We focused on the median effective dose (ED50) as an indicator of relative potency, calculating it with the EDcomp function. Significant differences in compound effects between cell lines were assessed using $t$-values and $p$-values obtained from EDcomp.

**Retrospective I-SPY2 analysis.** We obtained I-SPY2 clinical trial metadata and microarray characterizations of 988 patient transcriptomes

from the Gene Expression Omnibus (GEO) (GSE194040). We matched 90% of the recorded genes to our training dataset, mean-imputed the remaining 10% of genes, and standardized the data using $z$-score transformation. We then evaluated the alignment of I-SPY2 patient data with CCLE cell line training data across the first two principal components. Next, we predicted drug sensitivity for each patient across 32 concentrations (1E-3 μM–300 μM) in response to all 545 compounds and compound pairs in the CTRP. We generated patient-drug response predictions using independent models and computed the area under the curve (AUC) of each predicted dose-response assay. We scaled the AUC of each patient-drug prediction between 0-1 based on the drug's minimum and maximum predicted AUC across all I-SPY2 patients.

Participants in the I-SPY2 trial were assigned to treatment arms based on classification into one of eight subtypes, determined by HR, ERBB2-receptor, and MammaPrint status. Adaptive randomization was employed to dynamically adjust treatment assignments based on ongoing analysis of treatment outcomes, optimizing the likelihood of each patient achieving a pathological complete response. The trial assessed the efficacy of various combination therapies relative to paclitaxel treatment, the clinical standard of care. We identified drugs matched between I-SPY2 treatment arms and the CTRP, including paclitaxel, neratinib, MK2206, veliparib, and carboplatin. In the I-SPY2 experimental arms, patients were treated with a combination of paclitaxel and an additional drug(s) to assess response relative to paclitaxel treatment only. As the predictive ability of ChemProbe was only evaluated with respect to the available compounds and compound pairs in the CTRP, the ChemProbe predictions for I-SPY2 patients reflected predicted patient response to a single compound rather than a combination therapy.

**Prospective differential potency predictions.** To identify differentially potent compounds between HCC1806-Par and MDA-MB-231-Par cell lines, we computed the difference in predicted IC50 values for compounds that passed dose-response modeling. We visually examined dose-response curves of the top 50 differentially sensitive compounds and selected candidates for in vitro testing. We based selection criteria on the completeness of dose-response curves in each cell line, including adequate Emax and Emin boundaries within the predicted concentration range.

We conducted a preliminary dose-response experiment to determine appropriate concentration points for the subsequent dose-response experiments across a broader range of concentrations than our predictions (300—1.7e-3 μM, 12 points) (Supplementary Fig. 5b, c). We narrowed the concentration range for the following experiment to capture response granularity (100—2.1e-6 μM, 12 points) (Fig. 4).

**Integrated gradients.** We employed integrated gradients, a path-based model attribution technique, to determine the extent to which feature gradients changed compared to a baseline feature vector. The method involves linearly interpolating $n$ feature vectors between a designated baseline and the query feature vector. We used zero-vector baselines for compound and gene expression features and set $n = 50$ as the step size. At each interpolated feature vector step, gradients of the inputs are calculated with respect to the corresponding prediction. Finally, the integral of each feature along the path of feature gradients between the baseline vector and the query vector is computed. The Python package captum was used to compute integrated gradients.

To account for potential differences in cellular responses, we used the predicted compound IC50 for each cell line-compound pair to calculate integrated gradients and obtain an attribution vector at the predicted IC50. We then extracted the cell line feature attribution vector for each pair to investigate the influence of conditional compound information on gradient changes in the input gene expression features. To address cell line-specific effects, we standardized the attribution features of each cell line separately using a $z$-score transformation, resulting in adjusted attribution vectors (Supplementary Fig. 6b, c).

**Attribution method soundness checks.** To evaluate the sensitivity of the attribution method to learned parameters and data features, we conducted soundness checks. First, we assessed model-dependent attribution method invariance by comparing the attribution vectors of randomly initialized parameters of architecturally identical models with those of the trained models. We applied integrated gradients to the trained and randomly initialized models and compared the attribution vectors. Second, we evaluated data-dependent attribution method invariance by permuting the data labels, training architecturally identical models, and applying integrated gradients to compare the true-model and permuted-model attribution vectors (Supplementary Fig. 6c). We used the correlation between the attribution vectors of the true and alternative models to assess the attribution method's sensitivity to learned parameters and dependence between data features and labels.

**Attribution similarity analysis.** We investigated the relationship between compound MOAs and learned gene expression feature dependence by examining attribution vector similarity. First, we filtered attribution vectors by considering compound MOA classes with at least two compounds successfully attributed in all seven cell lines to obtain MOA classes with sufficient samples for analysis (control compound set). This resulted in 28 MOA classes, which served as a true label baseline. We then compared these true labels to unsupervised labels generated by $K$-means clustering of attribution vectors from a trained model, a randomly initialized model, compound fingerprints, and a label-permutation baseline (Fig. 5b). We applied $K$-means clustering on five independent trials.

We analyzed gene target attributions to further investigate the model dependence on individual nominal targets within each MOA class. Specifically, we applied a two-sided Wilcoxon rank-sum test to group attributions for each nominal target in the MOA class of interest and adjusted for false discovery rate (FDR) using the Benjamini-Hochberg (BH) procedure. We visualized the nominal target with the largest average attribution difference between groups for each MOA class (Fig. 5c).

**Attribution network analysis.** We extended our analysis to include all attribution vectors generated from the 7-cell line test set. We randomly selected a single nominal target from each compound MOA class to avoid bias towards closely associated targets. This is because the nominal targets of a single compound likely fall in close network proximity, and downstream network analysis of target sets would reflect artificial over-connectivity. For example, the MOA class of neratinib includes nominal targets EGFR and HER2, which are involved in the same pathway. Therefore, we randomly chose one target from this set.

We applied Leiden clustering unsupervised discovery of attribution clusters. As described above, we defined attribution cluster MOA classes by random target selection from each compound MOA class. We filtered the STRING database to consider only high-confidence protein–protein associations (combined score > 0.7). We queried STRING for attribution cluster nominal targets and computed the connectivity of the resulting subgraph. To account for random subgraph connectivity due to target biases in STRING, we randomly sampled from available targets, queried the filtered STRING database, and computed connectivity. We repeated this procedure with randomly sampled protein-coding genes to account for random protein associations (Fig. 5f). We used the Networkx library for analysis.

To test for protein interaction enrichment, we defined attribution cluster nominal targets by random target selection from each compound MOA class, as described above (number of targets > 3). Next, we queried the STRING API for protein–protein interaction enrichment in the network of high-confidence protein–protein associations (combined score > 0.7). We computed statistical enrichment using the hypergeometric test, which tests if a query set of proteins has more interactions than expected relative to the background proteome-wide interaction distribution. We also applied the hypergeometric test for functional enrichment of GO terms, KEGG pathways, and Reactome pathways. We used the stringdb python package to

access the STRING API. To infer potential modules of action for compounds, we selected the unique set of all nominal targets associated with an attribution cluster.

**Cell line characterization and differential expression analysis**. RNA sequencing was conducted on seven test cell lines in triplicate, namely HCC1806-Par, HCC1806-LMb/c, MDA-MB-231-Par, MDA-MB-231-LM2, SW480, and SW480-LvM2[56]. Using RNA that was rRNA-depleted with Ribo-Zero Gold (Illumina), libraries were prepared with SciptSeq-v2 (Illumina) and sequenced on an Illumina HiSeq4000 at UCSF Center for Advanced Technologies. Transcript abundances were quantified using Salmon, and tximport was utilized to summarize transcript-level measurements. We employed DESeq2 to identify differentially expressed genes ($n = 3$ per condition).

**Differential attribution analysis**. To assess model dependence on individual genes within attribution clusters, we conducted an unbiased analysis. We applied a two-sided Wilcoxon rank-sum test to each gene to analyze gene attributions within a cluster relative to all remaining samples. We adjusted for FDR using BH to account for multiple testing. We utilized Scanpy to apply tests across genes in each cluster relative to all other samples. Attributions were standard scaled and each cluster's top-10 most significant genes were plotted. Leiden groups were hierarchically clustered (complete linkage) by Pearson correlation. Scanpy was used for computation and visualization. We obtained gene expression—sensitivity Pearson correlation $z$-scores and corresponding visualizations from the Cancer Therapeutics Response Portal v2 feature correlation analysis (Supplementary Fig. 8b, c).

**Software and code reporting**
Data collection tools: Python (3.10.6) was used to collect data, along with the following packages: stringdb (0.1.5), scanpy (1.9.3). Data analysis tools: Python (3.10.6) was used along with the following packages: numpy (1.23.4), pandas (1.5.1), matplotlib (3.6.2), seaborn (0.11.2), scanpy (1.9.3), scipy (1.9.3), scikit-learn (1.1.3), statsmodels (0.13.5), rdkit (2022.9.4), pytorch (1.13.0), pytorch-lightning (1.8.4). R (4.0.2) was used along with the following packages: tidyverse (1.13.0), tximport (1.18.0), genomicfeatures (1.42.2), deseq2 (1.30.1), enhancedvolcano (1.8.0), drc (3.0_1).

**Life science study design**
Samples for prospective in vitro testing of model predictions were selected based on the presence of complete dose-response curves among the top 50 differential predictions for the cell line pair. The chosen predicted dose-response curves exhibited adequate $E_{max}$ and $E_{min}$ boundaries, appropriately covering the predicted concentration range. Given resource constraints, we reasoned that six out of 50 compounds (12%) provided adequate representativeness of model predictions.

In the prospective drug screening experiment (Fig. 4a–f; Supplementary Fig. 4 and Supplementary Data 7), one plate showed significant cell death in four wells within column 6. To maintain the integrity of the analysis and avoid distorting the representation of the data, the values from these four wells were excluded from the analysis. Specifically, two wells were used for testing the drug CAY10618 against the cell lines HCC1806-Par and MDA-MB-231-Par. Similarly, two wells were dedicated to testing the drug neratinib against the same cell lines, HCC1806-Par and MDA-MB-231-Par.

As outlined in the methods, we initially conducted a preliminary screen to calibrate the dose-response concentration ranges of the tested compounds (300—1.7e-3 µM, 12 points) (Supplementary Fig. 4b, c and Supplementary Data 7). Based on the insights gained from the preliminary screen, we refined the concentration range for the subsequent experiment to enhance the capture of response granularity (100—2.1e-6 µM, 12 points) (Fig. 4 and Supplementary Data 7). Both experiments consistently demonstrated a significant relationship between the responses of the differential compounds (Fig. 4g; Supplementary Fig. 4c and Supplementary Data 7).

In the context of our prospective experiments, biological sample randomization was not applicable. However, for model training and evaluation, we employed a numerical random split of samples by cell line into groups for five-fold cross-validation.

Blinding was deemed unnecessary for our study, as the prospective experiments were solely determined by the objective predictions of an algorithm.

Cell line sources: MDA-MB-231-Par (ATCC HTB-26); HCC1806-Par (ATCC CRL-2335).

**Use of large language models (LLMs)**
We used OpenAI ChatGPT 3.5 Turbo and ChatGPT 4 as scientific editing tools when writing the manuscript. We prompted the LLMs to suggest revisions to our manually drafted text for improved clarity and conciseness, predominantly at the paragraph level. We did not ask the LLMs to generate content de novo. An example of a prompt we used was, "You are helping edit papers for a broad scientific audience, emphasizing clarity and conciseness. Revise the following paragraph: <draft text here > ." We manually reviewed the LLMs' suggested revised text and decided whether to include part, all, or none of it on a word-by-word basis.

**Reporting summary**
Further information on research design is available in the Nature Portfolio Reporting Summary linked to this article.

## Data availability
Training and validation data from CTRP v1/2 can be downloaded at ftp://caftpd.nci.nih.gov/pub/OCG-DCC/CTD2/Broad/CTRPv2.0_2015_ctd2_ExpandedDataset/CTRPv2.0_2015_ctd2_ExpandedDataset.zip[27] CCLE expression data can be downloaded at https://ndownloader.figshare.com/files/24613325. CCLE sample metadata can be downloaded at https://ndownloader.figshare.com/files/24613394[57]. I-SPY2 gene expression data is located at GSE194040. I-SPY2 patient-level biomarker scores, subtype classes, and clinical/response data were gathered from supplementary information of Wolf, et al.: https://www.cell.com/cms/10.1016/j.ccell.2022.05.005/attachment/c220411b-c281-41e8-befa-a45e48af9c64/mmc3.xlsx[33] HGNC was used to map gene names: https://www.genenames.org/tools/multi-symbol-checker/. Protein–protein interaction data was downloaded from the STRING database v11.5. The current file version is found here: https://stringdb-downloads.org/download/protein.links.v12.0/9606.protein.links.v12.0.txt.gz. CCLE gene expression—sensitivity Pearson correlation $z$-scores and corresponding visualizations were obtained from the CTRP v1/2 web portal: https://portals.broadinstitute.org/ctrp.v2.1/. RNA sequencing gene expression profiles of triple-negative breast cancer cell line HCC1143 WT and LRP8 KO were obtained from a data access request to Zhipeng Li as original data from a related publication[49]. The Enrichr web portal was used to perform Wikipathway, KEGG, and GO enrichment analysis (https://maayanlab.cloud/Enrichr/). Source data for all tables and figures are provided in Supplementary Data.

## Code availability
Our code to download, preprocess data, reproduce model training, load pre-trained weights, and run model inference is available as open source at https://github.com/keiserlab/chemprobe under the MIT License[58].

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

## Acknowledgements

This work was supported by CZI grant DAF2018-191905 (https://doi.org/10.37921/550142lkcjzw) from the Chan Zuckerberg Initiative DAF, an advised fund of Silicon Valley Community Foundation (funder https://doi.org/10.13039/100014989) (M.J.K.), the National Institute of Mental Health of the National Institutes of Health (U01 MH115747-02 M.J.K.). We thank Zhipeng Li for his valuable insights into ferroptosis interpretation and for generously sharing the sequencing data of the LRP8 KO cell line.

## Author contributions

Conceptualization, W.C.; Methodology, W.C.; Software, W.C.; Validation, K.G.; Formal analysis, W.C.; Investigation, W.C., K.G.; Data curation, W.C.; Writing - Original Draft, W.C.; Writing - Review & Editing, W.C., M.J.K.; Visualization, W.C.; Supervision, H.G., M.J.K.; Project administration, M.J.K.; Funding acquisition, H.G., M.J.K.

## Competing interests

The authors declare no competing interests.
