## [Peer Review File · Communications Biology]

Reviewers' comments:

Reviewer #1 (Remarks to the Author):

The authors proposed a new deep learning method for studying the underexplored interactions between molecular profiles and chemical features for discovering biological mechanisms in drug response prediction models. Thereby, they demonstrated its properties extensively and comprehensively and furthermore showed solid external validation and intriguing biological/clinical results. Considering the comments and suggestions given below will strengthen the manuscript further.

Major:

Section 'ChemProbe predicts breast cancer patient response': The authors claim that their model achieves better accuracy than the I-SPY2 clinical trial. However, what served as a prediction from the I-SPY2 clinical trial? Did the treatment allocations serve as response predictors? That would also explain why 'Extended Data Fig. 2' shows zero patients predicted to be non-responders, i.e. no false negatives. Furthermore, did the authors use their adaptively randomized treatment allocations as predictions, or the reclassified subgroups? It seems to be a somehow unfair comparison, since identifying likely non-responsive patients is neat, however, if no better treatment is available, these patients will still be allocated to receive the same treatment. Potentially the authors could clarify this and potentially soften the claims.

Section 'ChemProbe predicts cellular drug sensitivity': In 'Fig 3c,d' only 2 out of the 6 chosen compounds and their prediction curves are shown. Is there any reason for this?

'On analysis, 50% of ModOA reflected significant network interaction enrichment and a variety of functional enrichments from gene ontologies, KEGG pathways, and Reactome pathways (Fig 5f).': The authors only show 3 examples, are these the most significant network interaction enrichments? It would be nice if this could be formulated in a more quantitative manner. Furthermore, the authors state functional enrichments (KEGG, GO), however, there is no data shown to support this claim. The remaining results remain impressive and sound without these additional claims. Thus, the suggestion is to either show the results or to remove the claims from the manuscript.

Section 'Gene expression attribution vector pass interpretability soundness checks': The authors aim their feature attribution vectors to not correlate with baseline gene expression. They describe this result as an unfavorable confounder, however, would it not be a rather favorable

property of the attribution to select highly expressed genes ? Are highly expressed genes not expected to be more predictive because of higher robustness ? The authors could further discuss this.

Minor:

In the introduction the authors present a complete and comprehensive overview over the discovery of drug response mechanisms from drug response prediction models. However, it would be nice to more highlight the limitations of current models and how their method builds on top of this, i.e. offers interpretability of interactions between molecular profiles and chemical structures without biological priors, in order to understand their contribution to the field.

It is not clear from the introduction how exactly the authors achieved interpretability of their drug response prediction model, i.e. integrated gradient feature attribution. Also, the fact that the authors used a 'conditional' deep learning model is missing from the introduction.

Method section 'Pharmacogenomic dataset': In the brackets '(Seashore-Ludow)', the authors most likely want to cite 10.1158/2159-8290.CD-15-0235

'ChemProbe+/- classification accuracy significantly outperformed I-SPY2 ($p < 5e-2$, Fig 2d)', probably this refers to 'Fig 2c'.

'Target modules had significantly greater connectivity than modules generated from randomly sampled target protein sets or randomly sampled protein sets (Fig 5d).', probably refers to 'Fig 5c'. Furthermore, the actual 'Fig. 5d' would have to be most likely referenced in the prior sentence.

Section 'Gene expression attribution vector pass interpretability soundness checks': The authors describe that in order to alleviate correlations between the attribution and baseline gene expression vectors, they 'normalized attribution vectors by line'. Do the authors mean "cell line" ? Does this mean feature-wise or sample-wise normalization ? What method was used (Z-scores, min-max etc.) ?

Typo in Fig 5 legend: '(f) Network representation of select clustered target profile subgraphs.' -> "selected"

Consider removing informal language such as 'Digging down', 'catastrophically' or 'are a moving target' from the manuscript

'To verify that these results were not artifacts of the transcriptomes or relative gene expression differences, we also performed differential expression analysis (DEA) between MDA-MB-231-Par and HCC1806-Par. Besides GPX4, a key ferroptosis regulator, no ferroptosis-associated genes rose to significance ($p < 5e-2$, Extended Data Fig 6a).' The authors demonstrate no associations between gene expression of ferroptosis regulators in the two cell lines. If understood correctly, the DEA analysis only contained two samples, which would explain this negative result. Why would it be unfavorable if ferroptosis regulators are differentially expressed? Is it not desirable to have a biomarker with a high effect size?

Reviewer #2 (Remarks to the Author):

The author developed ChemProbe, a model that predicts cellular sensitivity to hundreds of molecular probes and drugs by learning to combine transcriptomes and chemical structures. Generally, the question is important. The idea is straight forward. I have the following questions.

1. The author takes both chem and transcriptome as input for drug response prediction. Since the number of drugs might be much less than the number of transcriptomes. Although the input paired chem and transcriptome, in total there might be only a few chems but lots of transcriptomes. So, we may also model the question as a multiple-classification problem. Input the transcriptome, and predict which chem may respond or not. Could the authors explain the difference? and is there a difference in terms of results between these two ways for modeling.
2. the dimension of the transcriptome is more than 10000. how does the author perform embedding? Similar question to Chem. The introduction to the feature extraction part is too simple.
3. the block used in the algorithm is not novel or new. For example, the chem currently is usually modeled as a graph. Have the authors give it a try?
4. drug response prediction is not a new topic, but I did see that author compared their method with the Sota method. For example, Few-shot learning creates predictive models of drug response that translate from high-throughput screens to individual patients

Reviewer #3 co-reviewed with reviewer 1

Response to Reviewer Comments

We appreciate the Reviewers' support of the study and their thoughtful feedback to strengthen it. We have endeavored to adopt and respond to their recommendations by adding clarifying and motivating text to the manuscript as well as two new Supplementary Figures and a new table of Supplementary Data.

Reviewer 1

The authors proposed a new deep learning method for studying the underexplored interactions between molecular profiles and chemical features for discovering biological mechanisms in drug response prediction models. Thereby, they demonstrated its properties extensively and comprehensively and furthermore showed solid external validation and intriguing biological/clinical results. Considering the comments and suggestions given below will strengthen the manuscript further.

We thank the Reviewer for their support of the study and for their clarifying questions regarding the methodology and interpretation of results. We have adopted their feedback and revised the text as detailed below.

Major:

1. Section 'ChemProbe predicts breast cancer patient response': The authors claim that their model achieves better accuracy than the I-SPY2 clinical trial. However, what served as a prediction from the I-SPY2 clinical trial? Did the treatment allocations serve as response predictors? That would also explain why 'Extended Data Fig. 2' shows zero patients predicted to be non-responders, i.e. no false negatives. Furthermore, did the authors use their adaptively randomized treatment allocations as predictions, or the reclassified subgroups? It seems to be a somehow unfair comparison, since identifying likely non-responsive patients is neat, however, if no better treatment is available, these patients will still be allocated to receive the same treatment. Potentially the authors could clarify this and potentially soften the claims.

Per the Reviewer's request, we have clarified in the main text and methods that the original, adaptively randomized I-SPY-2 treatment allocations function as response predictors:

"We used gene expression and patient drug response data from the I-SPY2 adaptive, randomized, phase II clinical trial of neoadjuvant therapies for early-stage breast cancer (NCT01042379)^{32,33}. I-SPY2 assigned patients to treatment arms based on biomarkers such as hormone receptor status, human epidermal growth factor receptor-2 expression, and MammaPrint status."

"We compared ChemProbe predictions with the original treatment allocations in the I-SPY2 trial, which were determined by standard biomarkers of the participants' tumors."

"Participants in the I-SPY2 trial were assigned to treatment arms based on classification into one of eight subtypes, determined by HR, ERBB2-receptor, and MammaPrint status. Adaptive randomization was employed to dynamically adjust treatment assignments based on ongoing analysis of treatment outcomes, optimizing the likelihood of each patient achieving a pathological complete response."

Considering the significant burden of undergoing ineffective treatments and the common "trial and error" approach in managing chronic diseases, we emphasize the value of identifying patients unlikely to benefit from therapy. Wolf et al. (2022) propose alternative treatments based on reclassified subgroups using multiomic analyses. Their method involves a counterfactual approach, assuming these reclassified subgroups achieve a 100% success rate. This approach compares a theoretical pCR with actual outcomes. We considered this method too presumptive for the empirical reporting of results. We have included new discussion material to highlight this limitation:

“Although our data may suggest alternative drug treatments, we refrained from recommending therapies as their potential value is challenging to evaluate within the scope of this study.”

2. Section ‘ChemProbe predicts cellular drug sensitivity’: In ‘Fig 3c,d’ only 2 out of the 6 chosen compounds and their prediction curves are shown. Is there any reason for this ?

We originally included the curves for two of the six compounds to illustrate representative predicted dose-response curves and corresponding confidence intervals. However, we’re happy to include all plots in the Supplementary Information, in the new Supplementary Figure 4.

3. ‘On analysis, 50% of ModOA reflected significant network interaction enrichment and a variety of functional enrichments from gene ontologies, KEGG pathways, and Reactome pathways (Fig 5f).’: The authors only show 3 examples, are these the most significant network interaction enrichments ? It would be nice if this could be formulated in a more quantitative manner. Furthermore, the authors state functional enrichments (KEGG, GO), however, there is no data shown to support this claim. The remaining results remain impressive and sound without these additional claims. Thus, the suggestion is to either show the results or to remove the claims from the manuscript.

Supplementary Data 11 reports quantitative summary statistics for network interaction enrichment (sheet “Fig5f”). Following the Reviewer’s suggestion, we’ve added new visualizations of the additional networks as Supplementary Figure 7 and a new Supplementary Data 12 of functional enrichments (e.g., p-values and false discovery rates) as a standalone spreadsheet file.

4. Section ‘Gene expression attribution vector pass interpretability soundness checks’: The authors claim their feature attribution vectors to not correlate with baseline gene expression. They describe this result as an unfavorable confounder, however, would it not be a rather favorable property of the attribution to select highly expressed genes ? Are highly expressed genes not expected to be more predictive because of higher robustness ? The authors could further discuss this.

The Reviewer brings up an important tension between gene expression levels and their biological relevance. Classical methods leverage expression as a direct indicator of relevance, which is an effective starting point. However, not all highly expressed genes are relevant to a particular phenotypic outcome, and not all relevant genes are highly expressed. When an interpretability method identifies highly expressed genes, this is reassuring. However, which potentially relevant genes or gene combinations do expression magnitude-based prioritization overlook? These genes may be acting independently at average expression levels or may be interacting in concert with other genes to drive phenotypic effects. Consequently, we sought also to evaluate whether the model is learning biologically plausible feature dependencies that traditional methods fail to detect.

We designed our experiments to assess whether our model interpretation method, integrated gradients, functions independently of the particular model architecture and the data-generating process. The initial findings revealed that the attribution vectors indeed primarily reflected expression magnitude in the data unless we adjusted for cell line-specific effects. Consequently, at that stage, we could not reject the competing hypothesis that raw (unadjusted) attribution vectors represented data-specific artifacts rather than biologically informative interactions derived from the model’s feature learning. Ideally, a well-calibrated computational interpretation method should attribute significance to features based on their relevance to the model’s predictions (outputs) rather than their expression levels (inputs) alone.

Accordingly, in our subsequent experiments, we sought to empirically assess whether adjusted attributions aligned with established biological knowledge without depending on gene expression magnitude linearly determining the phenotypic outcome. For example, in our analysis of genes associated with ferroptosis, we found that whereas a conventional differential expression analysis (DEA) identified only one gene, our attribution-based method recognized all five relevant genes, thus providing a more comprehensive and potentially more accurate reflection of biological reality (refer to Figure 6c and Supplementary Figure 8a).

To clarify this motivation and balance in the text, we have added further context to the main text that motivates our method for validating the reliability of attributions:

“Although some gene expression magnitudes may linearly correlate with phenotypes, this framework does not capture other causal, nonlinear interactions within gene regulatory networks. A well-calibrated interpretation method should attribute significance to features based on their relevance to the model's predictions, rather than solely on their expression levels.”

Minor:

5. In the introduction the authors present a complete and comprehensive overview over the discovery of drug response mechanisms from drug response prediction models. However, it would be nice to more highlight the limitations of current models and how their method builds on top of this, i.e. offers interpretability of interactions between molecular profiles and chemical structures without biological priors, in order to understand their contribution to the field.

Please see point 6, where we address both questions.

6. It is not clear from the introduction how exactly the authors achieved interpretability of their drug response prediction model, i.e. integrated gradient feature attribution. Also, the fact that the authors used a ‘conditional’ deep learning model is missing from the introduction.

We appreciate this feedback and have revised the final paragraph of the introduction. In response, we've addressed the missing methodological specifics and clarified motivations:

“While previous research has addressed the tradeoffs among various input feature modalities, few studies explore how diverse feature sets are integrated. Moreover, little work has been done to explore whether well-performing models' features reflect expected biological relationships. Our work investigates methods for integrating biological and chemical features to improve drug sensitivity prediction and assesses the utility of model interpretation methods for advancing biological discovery.”

7. Method section ‘Pharmacogenomic dataset’: In the brackets ‘(Seashore-Ludow)’, the authors most likely want to cite 10.1158/2159-8290.CD-15-0235

Please see point 9, where we address both questions.

8. ‘ChemProbe+/- classification accuracy significantly outperformed I-SPY2 ($p < 5e-2$, Fig 2d)’, probably this refers to ‘Fig 2c’.

Please see point 9, where we address both questions.

9. ‘Target modules had significantly greater connectivity than modules generated from randomly sampled target protein sets or randomly sampled protein sets (Fig 5d).’, probably refers to ‘Fig 5c’. Furthermore, the actual ‘Fig. 5d’ would have to be most likely referenced in the prior sentence.

We appreciate notice of these typographical errors and have corrected citation formats and updated references to correspond to the correct figures.

10. Section ‘Gene expression attribution vector pass interpretability soundness checks’: The authors describe that in order to alleviate correlations between the attribution and baseline gene expression vectors, they ‘normalized attribution vectors by line’. Do the authors mean “cell line” ? Does this mean feature-wise or sample-wise normalization ? What method was used (Z-scores, min-max etc.) ?

We regret the ambiguity in our phrasing. We have clarified our methodology for adjusting attribution vectors, revising the main text to specify that the normalization was performed by "cell line." We normalized feature-wise, but only across the observations specific to each cell line. We now describe this procedure in the methods section as follows:

“To address cell line-specific effects, we standardized the attribution features of each cell line separately using a Z-score transformation, resulting in adjusted attribution vectors (Supplementary Fig. 7b,c).”

11. Typo in Fig 5 legend: ‘(f) Network representation of select clustered target profile subgraphs.’ -> “selected”

Thank you for pointing this out. The term "select" in the legend for Figure 5 was used intentionally as an adjective, meaning "chosen" or "exemplar," to describe the clustered target profile subgraphs. We have revised it to "exemplar" to avoid ambiguity. As suggested in comment 3, we have included the other clustered target profile subgraphs in a new Supplementary Figure 7.

12. Consider removing informal language such as ‘Digging down’, ‘catastrophically’ or ‘are a moving target’ from the manuscript

Thank you, we have removed these examples of informal language.

13. ‘To verify that these results were not artifacts of the transcriptomes or relative gene expression differences, we also performed differential expression analysis (DEA) between MDA-MB-231-Par and HCC1806-Par. Besides GPX4, a key ferroptosis regulator, no ferroptosis-associated genes rose to significance ($p < 5e-2$, Extended Data Fig 6a).’ The authors demonstrate no associations between gene expression of ferroptosis regulators in the two cell lines. If understood correctly, the DEA analysis only contained two samples, which would explain this negative result. Why would it be unfavorable if ferroptosis regulators are differentially expressed? Is it not desirable to have a biomarker with a high effect size?

We thank the Reviewer for raising these questions regarding the DEA in the manuscript. To clarify, DEA conducted between the MDA-MB-231-Par and HCC1806-Par cell lines included three replicates per condition, a standard sample size for testing independent features. We have revised the Methods section of the manuscript to clearly state that RNA-seq was performed in triplicate.

Regarding interpretation, in many ways, this relates directly to Question 4 above. This section of the Results compares conventional DEA to a new and more nuanced method we introduce, which we term differential attribution analysis (DAA). As the Reviewer notes, it would indeed be favorable if the ferroptosis regulators were differentially expressed. Our focus on this point, however, is to demonstrate the limitations of DEA alone, which only incompletely establishes this connection. GPX4, a key ferroptosis regulator, was differentially expressed and identified by DEA. However, the inability of DEA to also detect other mechanistically important ferroptosis-associated genes highlights the potential limitations of expression-based methods for comprehensive discovery. In contrast, our proposed DAA method successfully identifies more relevant ferroptosis-associated genes. This demonstrates that (1) DAA does not merely reflect findings revealed by conventional DEA, and (2) the results obtained from DAA are orthogonally informative based on established biological knowledge. Indeed, SCD and LRP8 are among the top-7 highest scoring genes in our DAA, further illustrating the robustness of this approach (Supplementary Fig. 8d).

Reviewer 2

The author developed ChemProbe, a model that predicts cellular sensitivity to hundreds of molecular probes and drugs by learning to combine transcriptomes and chemical structures. Generally, the question is important. The idea is straight forward. I have the following questions.

We thank the Reviewer for their support of the study’s impact and approach.

1. The author takes both chem and transcriptome as input for drug response prediction. Since the number of drugs might be much less than the number of transcriptomes. Although the input paired chem and transcriptome, in total there might be only a few chems but lots of transcriptomes. So, we may also model the question as a multiple-classification problem. Input the transcriptome, and predict which chem may respond or not. Could the authors explain the difference? and is there a difference in terms of results between these two ways for modeling.

The Reviewer brings up an interesting question of how the conditional model architecture design compares to a more basic design that would instead model the prediction as a multi-task multi-label classification problem, where a single transcriptome input predicts responses to multiple chemicals (as output tasks). The multitask model would have trade-offs in expressivity, interpretability, and adaptability.

Expressivity. Multi-task classification requires setting arbitrary thresholds for "response" vs. "non-response," limiting the model's ability to express continuous variations in drug response. This risks oversimplifying biological variations and reduces the diversity of training samples.

Pharmacodynamic expressivity. ChemProbe takes account of chemical concentration to predict full dose-response curves, providing key pharmacodynamic metrics such as potency (IC50/EC50), efficacy (Emax), therapeutic index, and Hill coefficient. This functionality would be lost in a multitask classification formulation.

Interpretation of chemical features. A multitask classification architecture would no longer ingest any chemical structure-based features for its calculations. Incorporating chemical features allows us to assess how the interplay between chemical similarity in structure and mechanism of action (MOA) affects transcriptomic features similarly, which helps validate the biological relevance of our predictions (as shown in Supplementary Fig. 1a,b and Fig 5b).

Model adaptability. Incorporating chemical features makes our model more flexible and adaptable to new data, accommodating novel chemical structures through fine-tuning or retraining. Further, as a counterfactual, our evaluations show that removing chemical structural features from the input significantly degrades model performance in our concentration-dependent framework (refer to Table 1).

We have included a brief acknowledgment of these motivations in the Discussion:

"ChemProbe's conditional architecture enables expressive sensitivity predictions across chemical concentrations, enhancing model interpretability and adaptability to new probes."

2. the dimension of the transcriptome is more than 10000. how does the author perform embedding? Similar question to Chem. The introduction to the feature extraction part is too simple.

We appreciate the Reviewer's point and have revised the Methods to address it. We use scaled expression values for all protein-coding genes as the transcriptome input ($n = 19,114$), and RDKit Morgan fingerprints ($n = 512$, radius = 2) as the chemical input (Methods).

Neither input feature type meaningfully encodes locality among features – each feature (e.g., gene or chemical structure) is independent of its neighboring features in the vector. Thus we do not use convolutions or other locality-aggregating computations. Instead, as is typical, we embed using classical fully connected (affine) neural network layers. Accordingly, we used a straightforward fully connected embedding approach with decreasing sequential layer sizes. We have added the following information to the Methods section:

"In our experiments and publicly available trained models, we use a transcriptome embedding block with layers of size [2048, 512, 256] and a compound embedding block with layers of size [256, 128] to project to an embedding of size $g = c = 128$. We used $n = 2$ FiLM layers in the final models."

3. the block used in the algorithm is not novel or new. For example, the chem currently is usually modeled as a graph. Have the authors give it a try?

While graph-based representations of chemical structures are a promising approach, which we indeed actively use in other of our studies, this study focuses on integrating biological and chemical data using a method suitable for the (1) scale of this dataset and (2) scope of this study. In particular:

We recognize that using graphs to represent chemical structures is a significant area of research interest and is increasingly prevalent across various modeling applications. However, in our hands, effective graph-based models typically require at least 80,000 chemical structure examples, whereas our study included only 500 molecules (doi:10.26434/chemrxiv-2024-hznhx).

Here, we have explored methods that effectively integrate biological and chemical input data. Our literature review identified a gap in research on integrating these modalities. We drew inspiration from advancements in multimodal data integration involving text and images, specifically the use of feature-wise linear modulation (FiLM), which has shown significant enhancements in visual question answering (VQA). For example, in VQA, a textual question is paired with an image to generate a relevant answer, such as "[Image] [Q] Is the red ball on top of the blue box? [A] Yes."

Future work may indeed benefit from exploring different and continually evolving chemical representations and embedding techniques within this integration framework, but such investigations were beyond our intended scope. Our study focuses on the novel application of FiLM to bridge the gap in integrating chemical and biological data effectively. Our experiments evaluating the performance of various model-based data integration approaches highlight our pursuit of this question, as does our interpretation of learned model parameters (Table 1, Supplementary Fig. 1a,b).

To further clarify the motivation of this study, we have amended the introduction with the following:

"While previous research has addressed the trade-offs among various input feature modalities, few studies explore how diverse feature sets are integrated."

4. drug response prediction is not a new topic, but I did see that author compared their method with the Sota method. For example, Few-shot learning creates predictive models of drug response that translate from high-throughput screens to individual patients

We agree there is a wealth of intriguing applications and changing technical directions to explore in accurately modeling cellular sensitivity to chemical perturbations. As mentioned in the prior question, we see this as a downstream opportunity and a strength of the overall approach, because many of these techniques could eventually be modularly integrated as alternative embedding or loss-function strategies into the framework we report.

REVIEWERS' COMMENTS:

Reviewer #1 (Remarks to the Author):

Thanks, all comments have been addressed.

Response to Reviewer Comments

Reviewer 1

Thanks, all comments have been addressed.

Great to hear - thank you.